# Application of High-Resolution Melting and DNA Barcoding for Discrimination and Taxonomy Definition of Rocket Salad (*Diplotaxis* spp.) Species

**DOI:** 10.3390/genes14081594

**Published:** 2023-08-06

**Authors:** Pasquale Tripodi

**Affiliations:** Research Centre for Vegetable and Ornamental Crops, Council for Agricultural Research and Economics (CREA), 84098 Pontecagnano Faiano, Italy; pasquale.tripodi@crea.gov.it; Tel.: +39-089-386-217

**Keywords:** *Diplotaxis* spp., plastid markers, nuclear markers, DNA sequencing, structure analysis, high resolution melting, phylogenesis

## Abstract

Nuclear and cytoplasmic DNA barcoding regions are useful for plant identification, breeding, and phylogenesis. In this study, the genetic diversity of 17 *Diplotaxis* species, was investigated with 5 barcode markers. The allelic variation was based on the sequences of chloroplast DNA markers including the spacer between *trnL* and *trnF* and *tRNA-Phe* gene (*trnL-F*), the rubisco (*rbcl*), the maturase K (*matk*), as well as the internal transcribed spacer (ITS) region of the nuclear ribosomal DNA. A highly polymorphic marker (HRM500) derived from a comparison of cytoplasmic genome sequences in Brassicaceae, was also included. Subsequently, a real-time PCR method coupled with HRM analysis was implemented to better resolve taxonomic relationships and identify assays suitable for species identification. Integration of the five barcode regions revealed a grouping of the species according to the common chromosomal set number. Clusters including species with n = 11 (*D. duveryrieriana* or *cretacea*, *D. tenuifolia*, *D. simplex* and *D. acris*), n = 8 (*D. ibicensis*, *D. brevisiliqua* and *D. ilorcitana*), and n = 9 (*D. brachycarpa*, *D. virgata*, *D. assurgens*, and *D. berthautii*) chromosomes were identified. Both phylogenetic analysis and the genetic structure of the collection identified *D. siifolia* as the most distant species. Previous studies emphasized this species’ extremely high glucosinolate content, particularly for glucobrassicin. High-resolution melting analysis showed specific curve patterns useful for the discrimination of the species, thus determining ITS1 as the best barcode for fingerprinting. Findings demonstrate that the approach used in this study is effective for taxa investigations and genetic diversity studies.

## 1. Introduction

The genus *Diplotaxis* DC. is a member of the large Brassicaceae family, which includes over 4000 species with a relevant diversity in terms of plant architecture, leaf morphology, and content of nutraceutical compounds [1,2]. The genus includes over 30 species that originated from two main areas, including the Mediterranean basin, with a high level of endemic diversity in the northwest part of Africa and the Iberian Peninsula [3], and the western and southern Asian countries (mostly Turkey, Pakistan, and India) [3,4]. All species are diploid with different gametic chromosome numbers (7, 8, 9, 10, 11, and 13), except *D. muralis* (L.) DC, which is the only tetraploid (n = 21) derived from the hybridization of *D. tenuifolia* (L.) DC (n = 11) and *D. viminea* (L.) DC (n = 10) [5]. Since antiquity, *Diplotaxis* species have been widely used as a food and non-food commodity, such as oil, deodorant, cosmetic, and for medicinal purposes due to their inflammatory and depurative effects. Among them, *D. tenuifolia* (L.) DC, known as wild or perennial rocket, is the most relevant for alimentary uses and it is consumed worldwide as a leafy vegetable in mixing ready-to-use salads. Although the other species are not recognized for economic importance, they are reported to encompass a discrete level of diversity for the content of glucosinolates [6]. So far, the relationship within the taxa has been investigated through different approaches, including morphological assessments [7,8], biochemical studies [9,10], random DNA marker assays such as inter simple sequence repeat (ISSR) [4], and random amplification of polymorphic DNA (RAPD) [11]. These studies demonstrated the existence of two major clusters that partially distribute the accessions according to chromosome number. 

A first branch classifies accessions with n = 11 (*D. tenuifolia*, *D. cretacea*, and *D. simplex*), n = 10 (*D. viminea*), and the derived allopolyploid (*D. muralis*), while a second branch classifies the remaining accessions according to a diverse degree of relationship. Chloroplast and nuclear DNA markers showed a division of the genus into the *Brassica rapa*/*oleoracea* and *Brassica nigra* lineages [12,13]. The close relationship makes these species compatible for hybridization, thus suitable for the genetic improvement of cultivated brassicas. Although in the Brassicaceae family, the systematics of *Diplotaxis* spp. in relation to close taxa have been clearly determined, inconsistencies still occur for species relationships internal to the genus. These are probably due to the typology of markers so far used as well as the method used for polymorphism detection based on gel-electrophoresis techniques. Therefore, the phylogeny is still not well-defined and no further advances for a better definition of taxonomic relationships through genetic investigation have been reported over the past 15 years.

Amplification of highly conserved sequences in plants represents a powerful strategy for fingerprinting and taxonomy objectives [14]. DNA barcoding is a flexible and highly accurate tool for species authentication. Barcode regions mostly involve the chloroplast genome that is uniparentally inherited from the maternal side, which has a very low rate of mutation and is unaffected by hybridization events [15]. The most promising markers designed on the chloroplast genome include the maturase K gene *matK* [16], the non-coding *trnL-F* intergenic spacer, and the Rubisco ribulose biphosphate carboxylase large chain *rbcl* gene [17]. Furthermore, the internal transcribed spacer ITS of nuclear ribosomal DNA (18S-26S) is a widely employed region for taxonomic research [18]. These genes are accepted by the community as the universal standards for species identification and molecular systematics [19,20], being extensively applied in the plant kingdom thanks to the easiness of amplification and sequencing [21]. Sequencing of barcode regions ensures the identification of single nucleotide polymorphisms (SNPs) that underlie the molecular signature of species taxa [22].

Implementing high-resolution melting (HRM) methods based on quantitative polymerase chain reaction (qPCR) coupled with melting curve analysis provides a fast and highly sensitive approach to genotyping [23]. HRM relies on the detection of mutations in the target sequence of amplicons based on dissociation curves generated during DNA denaturation [24]. Several studies combine HRM technology with DNA barcodes for plant authentication with high accuracy [25,26,27,28]. Therefore, HRM offers a valid method to be applied for the detection of genetic diversity in crops toward species identification and phylogenesis aims.

Here is reported the investigation of genetic variability and phylogenetic relationships of the *Diplotaxis* genus at the intraspecific level, analyzing both chloroplast and nuclear barcode DNA regions. Hence, it provides an effective and rapid method to identify *Diplotaxis* species and detect polymorphisms through combined sequencing and HRM approaches. This study represents a first attempt to investigate *Diplotaxis* species complementing barcode marker sequencing and quantitative PCR high-resolution melting.

## 2. Materials and Methods

### 2.1. Plant Material

The plant material consisted of 17 *Diplotaxis* species retrieved from different European genebanks (Figure 1) including the Leibniz-Institut für Pflanzengenetik und Kulturpflanzenforschung, IPK (Gatersleben, Germany), the Royal Botanical Garden Kew (Richmond, United Kingdom), the Universidad Politécnica de Madrid UPM (Madrid, Spain), and the Universidad de Castilla—La Mancha UCLM (Ciudad Real, Spain). 

Seeds were sown in pots under climate-grown chamber conditions at the Research Centre for Vegetable and Ornamental Crops (Italy). After 30 days from germination, 100 mg of fresh leaves were collected, followed by nucleic acid isolation using the DNeasy plant mini kit (Qiagen, Hilden, Germany). The DNA purity was estimated by absorbance at 280 and 260 nm, respectively, using a UV-Vis Nanodrop (Thermo Scientific, Wilmington, DE, USA), and the integrity by electrophoresis on a 1.0% agarose gel. Concentration was measured using the Qubit 3 Fluorometer (Thermofisher, Waltham, MA, USA). A volume of 10 μL of extracted DNA, as well as the standards, were diluted in 190 μL of buffer prepared using 1 ul of dsDNA BR Reagent 200 × and 199 μL of dsDNA BR buffer furnished with a Qubit^®^ dsDNA BR assay kit (Thermofisher, Waltham, MA, USA). DNA was diluted at a working concentration (15 ng/μL) and then stored at −20 °C prior to analysis.

### 2.2. DNA Barcode Primer Design

Specific regions for DNA barcoding were designed on *Diplotaxis tenuifolia* sequences deposited in the nucleotide database at the NCBI (National Center for Biotechnology Information), including the 556 bp nucleotide sequence of the internal transcribed spacer 1, (ITS1) 5.8S nuclear ribosomal RNA gene (Genbank: EF601913.1), and 3 chloroplast sequences as following: (i) The 357 bp nucleotide sequence of the intergenic spacer *trnL-trnF* and *tRNA-Phe(trnF*) gene (Genbank: DQ984109.1), (ii) The 583 bp nucleotide sequence of the partial *rbcL* (ribulose-1,5-bisphosphate carboxylase/oxygenase large subunit) gene for RuBisCo (Genbank: HE963454.1), (iii) The 708 bp nucleotide sequence of the partial *matK* gene for maturase K (Genbank: HE967405.1). Primers were designed using Primer 3.0 (https://primer3.ut.ee/), ensuring product sizes ranging from 100 to 267 base pairs. In addition, the HRM500 designed on cpDNA polymorphic sites of brassica was included [29]. Marker details are in Table 1**.**

### 2.3. Amplification and Sequencing

For each barcode marker, DNA amplification was performed in 15 μL reactions containing 30 ng of template DNA, 2.0 pmol of each forward and reverse primer, 0.2 mM of each dNTP, and 1.0 U of high-fidelity *PFU Taq* DNA polymerase (Promega, Madison, WI, USA). A C-1000 Touch^TM^ thermal cycler (Bio-Rad, Hercules, CA, USA) was used for nucleic acid amplification. The PCR cycle was performed as follows: one cycle at 95 °C for 1 min; 35 cycles at 95 °C for 30 s, 58 °C for 45 s, and 72 °C for 1 min; one cycle of final extension of 72 °C for 3 min; and soak at 12 °C. Amplification products were visualized on 1% agarose (Lonza, USA) gels in buffer TBE (EDTA 2 mM, Tris base 89 mM, Boric acid 89 mM) and molecular size was assessed with 1 Kb Plus DNA ladder (Life TechnologiesTM, Carlsbad, CA, USA). The visualization of PCR products was performed by staining agarose gel with SYBR^®^ safe (Life TechnologiesTM), and the fluorescence was viewed using Gel DocTM XR (Biorad). Amplicons were then purified with ExoSAP-IT^tm^ (Thermofisher, Waltham, MA, USA). The sequencing reaction was prepared with a Big Dye Terminator v3.1 Cycle sequencing kit (Thermofisher, Foster City, CA, USA). The sequencing cycle consisted of a cycle of denaturation (96 °C, 1 min) and 25 cycles of amplification (96 °C, 10 s; 50 °C, 5 s; 60 °C, 2 min). Sequencing reactions were then purified using the X-Terminator Purification Kit. The SeqStudio™ Genetic Analyzer (Thermofisher, Waltham, MA, USA) was used for Sanger sequencing. SeqScape^®^ v2.0 (Thermofisher, Waltham, MA, USA) was used for base calling.

### 2.4. High-Resolution Melting Analysis

Real-time PCR was performed in 10 μL of total reaction volume containing 37.5 nanograms of genomic DNA (2.5 μL with conc of 15 ng/μL), 1× Precision Melt Supermix (Bio-Rad, Inc., Hercules, CA, USA) (5 μL of 2× Supermix), and 0.4 μL of unlabeled forward and reverse primers (final concentration 200 nM). For each sample, analysis was performed in triplicate, and a negative control was included. The assay was performed on a CFX 96 RealTime PCR System (Bio-Rad, Inc., Hercules, CA, USA) using the following protocol: 95 °C for 2 min by 40 cycles at 95 °C for 15 s and 58 °C for 30 s. The melting curve was obtained with an initial step at 95 °C for 30 s (heteroduplex formation) and 64 °C for 3 min, then rising of temperature from 65 °C to 95 °C with increments of 0.2 °C/cycle every 10 s. Fluorescence data were analyzed with the Precision Melt Analysis™ Software (Bio-Rad, Inc., Hercules, CA, USA).

The generated melting profiles allowed the discrimination of genotypes by assignment to specific clusters according to the different temperatures of melting. A percent of confidence to estimate that a given sample was properly assigned with the cluster was calculated. Values above 95% were considered as the threshold for properly assigned samples to specific clusters.

### 2.5. Phylogenetic Tree Data Analysis

Single sequences were trimmed for low quality and manually edited using Chromas Lite. The sequences were aligned with the CLUSTAL W program implemented in MEGA X using the default settings [30]. Aligned sequences were trimmed to the same size by removing any sequence gaps and unaligned ends. All trimmed sequences are in Appendix A. A phylogenetic tree was drawn with the Kimura two-model implemented in the neighbor-joining method. A total of 10,000 bootstraps were considered. Sequences were then concatenated, re-aligned, and trimmed to obtain a concatenated tree following the method and model above described. Pairwise genetic distances (p-distance model) between all generated sequences were calculated between the 17 *Diplotaxis* species. For HRM profiles, melting curves were transformed into binary data prior to analyses. Analyses were conducted in MEGA X software.

### 2.6. Genetic Diversity and Population Structure

Analysis of genetic structure was performed with the Bayesian clustering method implemented in STRUCTURE v.2.4 [31]. The admixture model and MCMC (Markov chain Monte Carlo) method for allele frequency calculation and detection of the best number of population (K) were used. Runs were performed using 50,000 MCMC iterations and 50,000 burn-in cycles, with the number of K ranging between 1 and 15, with 5 independent runs. The optimal numbers of subpopulations were determined according to Evanno’s test implemented in Structure Harvester [32]. A membership coefficient (qi) ≥ 0.50 was considered to infer individuals to a specific subpopulation. Accessions with values lower than 0.5 at each assigned K were considered as admixed. Principal component analysis was conducted in R by the function *prcomp* (package stats), and the biplot was drawn using the ggplot2 R package [33]. 

## 3. Results

### 3.1. Sequence Comparisons

A total of 85 sequences were produced (Appendix A). Sequence length and nucleotide percentage for the five considered barcode regions are reported in Table 2. HRM500 was the longest sequence, consisting of 322 aligned sites and 26 variant sites. Sequence length ranged from 309 (*D. viminea*) to 316 (*D. tenuisiliqua*) nucleotides. The nucleotide composition of HRM500 was A/T rich (34.27% A, 15.38% C, 14.44% G, 35.88% T). ITS1 consisted of 274 aligned sites and a larger number of polymorphic sites (163). Sequence length ranged from 214 (*D. acris*, *D. ilorcitana*, and *D. simplex*) to 264 (*D. siifolia*) nucleotides with a balanced A/T-G/C composition (30.37% A, 23.82% C, 24.84% G, 20.84% T). Of the 241 bp of total aligned nucleotides for *matk*, 46 were polymorphic. A minimum sequence length was observed in *D. acris* and *D. virgata* (233 bp), while the maximum length was in *D. muralis* (239 bp). The nucleotide makeup of the *matk* region was A/T rich, with 32.71% A, 12.71% C, 16.19% G, and 38.35% T. Both *rbcl* and *trnL-F* were the smallest analyzed regions with a total of 73 bp and 80 bp of aligned sequence, respectively.

The former showed a slightly higher G/C average content (20.32% A, 27.19% C, 25.02% G, 27.47% T), whereas the latter a higher A/T (30.51% A, 20.60% C, 15.53% G, 32.76% T). A total of 9 polymorphic sites were observed for *rbcl*, while *trnL-F* exhibited a high variant rate being 97.5%. After trimming, 304 nucleotides were retained for HRM500, 200 nucleotides for ITS1, and 231 nucleotides for *matk*, whereas for *rbcl* and *trnL-F*, a total of 65 and 60 nucleotides were kept, respectively. By merging, realigning, and trimming all sequences of the 5 barcode regions, a total of 857 nucleotides were obtained and used for the consensus concatenated phylogenetic tree (Figure 2). In total, 158 base substitution characters were obtained. The concatenated sequence was A/T rich, comprising on average 31.39% A, 31.97 T, 17.96 C, and 18.67 G.

The overall pairwise genetic distance was 0.034 among all *Diplotaxis* species, with values between accessions ranging from 0.006 (*D. tenuifolia-D. duveyrieriana*;) to 0.103 (*D. siifolia- D. berthautii*) Table 3. The highest values were observed between *D. siifolia* and the rest of the species (average p-distance = 0.087), whereas *D. ilorcitana* showed the lowest values (average p-distance = 0.024).

### 3.2. High-Resolution Melting Profiles

Polymorphisms among the 17 *Diplotaxis* species were detected based on the pattern derived from the normalized melt curve and derived difference curves (Figure 3). In total, 26 HRM profiles were revealed by the 5 barcode markers used, ranging from 2 (*rbcl*) to 9 (ITS1). HRM500 (Figure 3a) showed 5 melting curves: 2 grouping 10 and 4 *Diplotaxis* species, respectively, whereas 3 were specific for *D. siifolia*, *D. brachycarpa,* and *D. simplex*. A higher number of melting curves was shown by ITS1 (Figure 3b), which clearly discriminated 7 out of the 17 studied species including *D. brachycarpa*, *D. berthautii*, *D. eruicoides*, *D. siifolia*, *D. assurgens*, *D. virgata*, and *D. tenuisiliqua*. *Matk* (Figure 3c) evidenced 5 distinct melting curves that grouped from 2 to 7 species, except for *D. harra,* which showed a singular pattern. *Rbcl* (Figure 3d) instead showed two different melting profiles, distinguishing *D. duveyrieriana*, *D. simplex*, *D. tenuifolia,* and *D. tenuisiliqua* from the rest. Finally, *trnL-F* (Figure 3e) showed 5 melting curves with a major group, including 13 species and 4 profiles discriminating *D. tenuisiliqua*, *D. eruicoides*, *D. brachycarpa,* and *D. assurgens*.

Overall, among the five barcode regions, ITS1 was especially noteworthy for the identification of *Diplotaxis* species, whereas the other markers showed less specific melting curves for the considered species. In all instances, *D. ibicensis*, *D. ilorcitana,* and *D. virgata* showed the same curve patterns, whereas *D. tenuifolia* and *D. duveyrieriana* showed the same melting profile for all markers except ITS1. Singular patterns were instead found for *D. brachycarpa* using HRM500, ITS1, and *trnL-F*. 

### 3.3. Phylogenetic Analysis

Phylogenetic trees using separate matrices are shown in Figure 4. 

Although different relationships were resolved in each tree, clustering of *D. tenuifolia*, *D. duveyrieriana,* and *D. simplex* was observed in all instances. *D. muralis* clustered close to *D. viminea* with *matk* and *rbcl*, while HRM500, ITS1, and *trnL-F* grouped *D. muralis* with *D. brevisiliqua*, *D. acris,* and *D. eruicoides*, respectively. *D. brevisiliqua* clustered close to *D. ilorcitana* with *matk*, *rbcl,* and *trnL-F*, and close to *D. harra* with ITS1. Overall, ITS1, *rbcl,* and *trnL-F* divided the accessions into main groups, whereas HRM500 and *matk* highlighted a higher number of specific subgroups. For a better resolution of the diversity, the individual sequences of the 17 accessions for the 5 markers were combined. The consensus tree (Figure 5a) separated *D. siifolia* from the rest and clustered the remaining accessions in 2 main groups including 11 and 5 accessions, respectively, defining furthermore several subclusters. The cluster of *D. tenuifolia*, *D. duveyrieriana,* and *D. simplex* was confirmed, whereas close relationships were found between *D. muralis* and *D. viminea* as well as *D. brevisiliqua*, *D. ilorcitana,* and *D. ibicensis*. A second group comprised the accessions *D. eruicoides*, *D. brachicarpa*, *D. virgata*, *D. assurgens,* and *D. berthautii,* with the latter two species showing a high level of similarity. The phylogenetic tree drawn using high-resolution melting analysis (Figure 5b) mostly confirmed the two main clusters observed by sequence alignments with few differences related to *D. virgata,* which was positioned separately in the larger group, and *D. siifolia,* which clustered within the second group, although tended to be as an outgroup accession. The high degree of similarity between *D. ibicensis*/*D ilorcitana*, *D. muralis*/*D. viminea*, *D. tenuifolia*/*D. duveyrieriana* was confirmed.

### 3.4. Population Structure

Polymorphisms were based on STRUCTURE analysis (Figure 6a), and the collection was divided into K = 2 as the likely number of subpopulations according to Evanno’s test (Figure 6b). The main group comprised 16 accessions, of which 8 (*D. ibicensis*, *D. duveyrieriana*, *D. simplex*, *D. tenuifolia*, *D. acris*, *D. muralis*, *D ilorcitana,* and *D. viminea*) had a high coefficient of membership (qi ≥ 0.9). *D. berthautii* and *D. siifolia* clustered in the second subpopulation with a coefficient of membership (qi) of 0.73 and 0.97, respectively. The principal component analysis grouped the accessions in both positive and negative axis of the first (PC_1_) and second (PC_2_) components. Accessions were mostly separated along the first component with 2 main groups comprising 11 and 5 genotypes positioned in the negative and positive axis of PC_1_, respectively. Different clusters with tightly close accessions were identified. The biggest ones included four (cluster A: *D. tenuifolia*, *D. duveyrieriana*, *D. simplex,* and *D. harra*) and five species (cluster B: *D. viminea*, *D. acris*, *D. ibicensis*, *D. muralis*, and *D. ilorcitana*), respectively. Two additional clusters, each comprising two species, were defined (cluster C: *D. tenuisiliqua* and *D. eruicoides*; cluster D: *D. virgata* and *D. brachycarpa*). *D. berthautii* and *D. siifolia* were confirmed to be highly diverse from the rest, being both located in the extreme parts of both PC_1_ and PC_2_.

## 4. Discussion

Molecular markers have innumerable uses being effectively employed to investigate the diversity in plant species, determine their phylogenetic relationships, or serve as selection markers in plant breeding [34]. Among these, marker assays targeting barcode regions offer a promising tool for species identification given the possibility to select standard loci to analyze taxonomically different specimens, thus producing comparable data [15]. The barcodes used in this study have been demonstrated to be powerful for species discrimination in diverse organisms including plants [28,35], mammals [36], fungi [37], and bacteria [38]. Furthermore, the application of HRM analysis coupled with DNA barcode has shown its potential for crop species identification [39,40].

Based on morphological characteristics, Prantl [41] and Schulz [42] placed the *Diplotaxis* species assayed in the present study into three subsections: Anocarpum, including *D. tenuifolia*, *D. simplex*, *D. viminea,* and *D. muralis*; Rynchocarpum, including *D. virgata,* and *D. erucoides*; and Catocarpum, including *D. harra*, *D. tenuifolia*, and *D. cretacea*. Gomez-Campo and Martínez-Laborde [43] subdivided Rhyncocarpum into three subclades: i) Rhynchocarpum, including *D. assurgens*, *D. berthautii*, *D. brachycarpa*, *D. siifolia*, *D. tenuisiliqua,* and *D. virgata*; ii) Heterocarpum, including *D. brevisiliqua*, *D. ibicensis,* and *D. ilorcitana*; and iii) Heteropetalum, containing *D. eruicoides*. Based on plastid DNA variation, Warwick et al. [12] placed these taxa into two different lineages of the subtribe *Brassicinae* (Brassicaceae family): the Rapa/Oleoracea combined seven species in three groups including *D. eruicoides* (Group A), *D. tenuifolia*, *D. duveryrieriana*, *D. simplex,* and *D. harra* (Group B), and *D. viminea* and *D. muralis* (Group C); the Nigra lineage included seven species in three subgroups: *D. brevisiliqua* and *D. ibicensis* (Group D), *D. brachycarpa* (Group E), and *D. tenuisiliqua*, *D. virgata*, *D. siifolia*, and *D. berthautii* (Group F). 

This study showed a grouping of species according to the common chromosomal set number in agreement with previous investigations based on cytological [5], cross compatibility [44], morphological [43], and molecular [4,11] approaches. The species with 11 chromosomes (n = 11) (*D. duveryrieriana* or *cretacea*, *D. tenuifolia*, *D. simplex,* and *D. acris*) constituted a group of closely related taxa that clustered in the same group with *D. viminea* (n = 10) and the amphidiploid *D. muralis* (n = 21), derived by the cross of *D. viminea* × *D. tenuifolia*. The close relationship of *D. muralis* with *D. viminea* rather than *D. tenuifolia* confirmed previous studies. The former line has been demonstrated to be the female parent of the amphidiploid from the peptide structure of the Rubisco enzyme and analysis with chloroplast markers [11,12,45]. The significant degree of resemblance between *D. tenuifolia* and *D. duveryriana* corroborated earlier findings reporting these two species belonging to a single cluster, with *D. simplex* joining a part [4]. Furthermore, analysis with ISSR highlighted a high level of similarity between *D. viminea* and *D. muralis* compared to the other species with n = 11 chromosomes [4]. From the biochemical point of view, this homogeneous complex has been reported to have a low total glucosinolate content and profiles of main glucosinolate compounds not dominated by any specific components [9]. The other group that includes the 3 species with 8 chromosomes (*D. ibicensis*, *D. brevisiliqua,* and *D. ilorcitana*) has confirmed previous investigations using RAPD markers [11], which placed *D. brevisiliqua* and *D. ilorcitana* in the same sub-cluster, whereas *D. ibicensis* was on another branch. These accessions were singled out by D’Antuono et al. [9] for their medium-high overall glucosinolate content, which was particularly high in sinigrin (allyl-glucosinolate). Beyond these two groupings, past investigations did not adequately resolve the remaining species under investigation. Four species with nine chromosomes, including *D. brachycarpa*, *D. virgata*, *D. assurgens*, and *D. berthautii*, were found in a primary cluster with *D. eruicoides*, whereas *D. tenuisiliqua* was on a different branch. *D. virgata* and *D. brachycarpa* were instead separated from *D. assurgens*, which was combined with *D. tenuisiliqua* in a study using RAPD markers [11]. Additionally, *D. eruicoides* was clustered on a separate branch in disaccord with the lineage-based classification [12], which grouped this species with taxa having 11 chromosomes. Considering biochemical profiles, *D. berthautii*, *D. tenuisiliqua,* and *D. virgata* were characterized by a high content of sinigrin, with *D. virgata* being also rich in gluconapin [9]. Based on sequence variation, *D. siifolia* was the most distant species from the rest, being positioned on a separated branch. This trend was confirmed by population structure analysis. However, the high-resolution melting-based dendrogram partially agreed with both the Rhynchocarpum classification proposed by Gomez-Campo and Martínez-Laborde [8] as well as the findings of Eschmann-Grupe and collaborators [11], which included this species in a cluster with *D. tenuisiliqua* and *D. assurgens*. D’Antuono et al. [9] indeed highlighted the very high glucosinolate content of *D. siifolia* in particular for the glucobrassicin compound. In agreement with the metabolic profile and considering the leaf morphology diversity compared to the other *Diplotaxis* species, barcode DNA fingerprinting supports the high diversity of *D. siifolia* compared to the rest.

Both DNA barcode and HRM analysis highlighted the similar relationships between species with the same chromosomal number, thus providing insight into the possibility of successful interspecific crosses. Beyond taxonomical investigation, HRM provides a reliable and cost-effective method for rapid identification of *Diplotaxis* species. The discrimination is crucial when the constituent species are in processed form and/or in mixed packages where alien product detection is difficult. Barcoding through HRM enables the detection of adulterants in admixed samples at very low concentrations, since it relies on sensitive melting curve changes through the release of a saturated intercalating dye from DNA duplex denaturation following the raising of the temperature [46]. In the present study, ITS1 revealed a greater rate of percent variation compared to the other barcode regions, thus allowing for distinguishing a higher number of species. These results agree with the larger discrimination power at the low taxonomic level of the nuclear internal transcribed region compared to the plastid regions [15,47]. Raising of availability of genomic resources will benefit the discovery of novel SNP across the genome, boosting the development of suitable HRM markers for genetic fingerprinting.

## 5. Conclusions

This study represents the first attempt to investigate *Diplotaxis* species with newly developed barcode markers which combine DNA sequencing and high-resolution melting analysis. The phylogenetic relationships among the 17 assayed species confirmed previous inferences from morphological, biochemical, and molecular data, thus indicating the reliability of the marker tested. Results better resolved the evolutionary distance of D. siifolia within the *Diplotaxis* gene pool. ITS1 has been found as the barcoding sequence with the highest discriminatory power for species identification. The DNA barcode and HRM confirm the effectiveness of the strategy utilized in this work for identifying species and examining genetic relationships.

## Figures and Tables

**Figure 1 genes-14-01594-f001:**
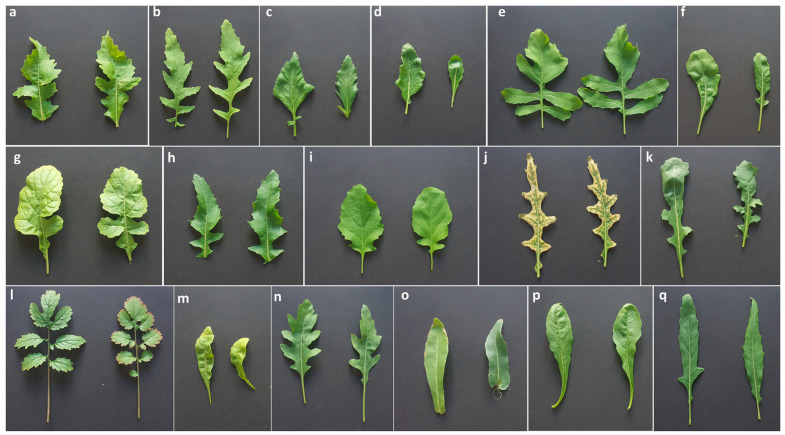
Plant material. (**a**) *Diplotaxis acris* (acc. 554488); (**b**) *Diplotaxis assurgens* (Delile) Thell. (acc. S17021002); (**c**) *Diplotaxis berthautii* Braun-Blanq. and Maire (acc. S17021001); (**d**) *Diplotaxis brachycarpa* (acc. S17021009); (**e**) *Diplotaxis brevisiliqua* (Coss.) Mart.-Laborde (acc. S17021010); (**f**) *Diplotaxis duveyrieriana* Coss. (cretacea) (acc. S17021008); (**g**) *Diplotaxis eruicoides* (L.) DC. (acc. DIPLO2); (**h**) *Diplotaxis harra* (acc. 338167); (**i**) *Diplotaxis ibicensis* (Pau) Gómez-Campo (acc. S17021006); (**j**) *Diplotaxis ilorcitana* (acc. S17021007); (**k**) *Diplotaxis muralis* (L.) DC. (acc. DIPLO 5); (**l**) *Diplotaxis siifolia* Kunze (acc. S17021003); (**m**) *Diplotaxis simplex* (Viv.) Spreng. (acc. S17021004); (**n**) *Diplotaxis tenuifolia* (L.) DC. (acc. DIPLO 12); (**o**) *Diplotaxis tenuisiliqua* (acc. DIPLO 10); (**p**) *Diplotaxis viminea* (L.) DC. (acc. S17021005); (**q**) *Diplotaxis virgata* (Cav.) DC (acc. CM 0025909). The Royal Botanical Garden Kew (Richmond, United Kingdom) provided (**a**,**h**). The Universidad Politécnica de Madrid UPM (Madrid, Spain) provided (**b**–**f**,**i**,**j**,**l**,**m**,**p**). The Leibniz-Institut für Pflanzengenetik und Kulturpflanzenforschung, IPK (Gatersleben, Germany) provided (**g**,**k**,**n**,**o**). The Universidad de Castilla—La Mancha UCLM (Ciudad Real, Spain) provided (**q**). All images are on the same size scale.

**Figure 2 genes-14-01594-f002:**
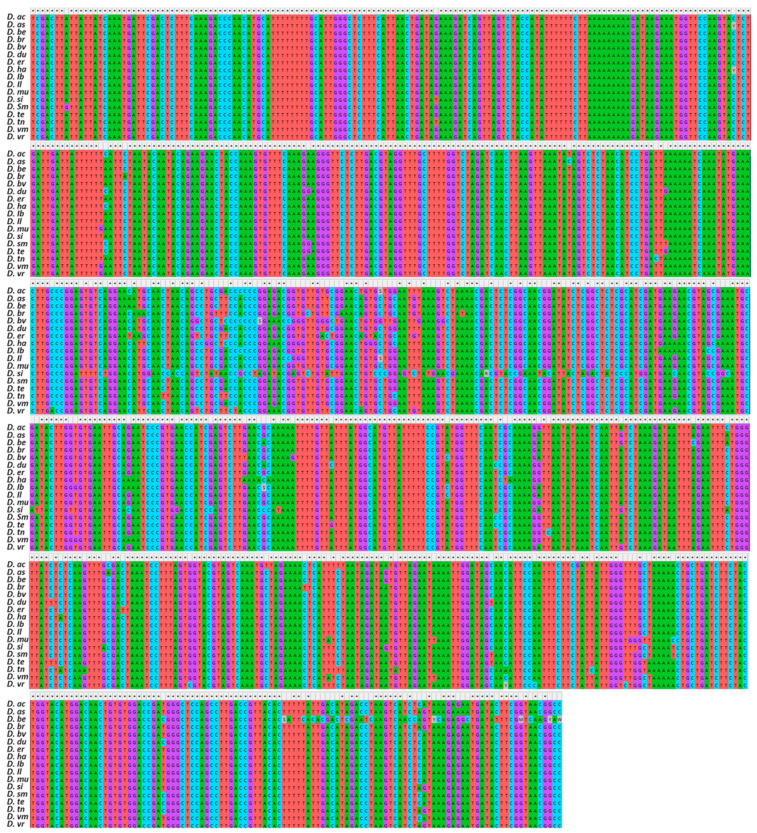
DNA sequence alignment analysis of concatenated ITS1, *trnL-F*, *rbcl*, *matk*, and HRM500 sequences for the *Diplotaxis* species considered in the present study. For acronyms, see Table 2 caption.

**Figure 3 genes-14-01594-f003:**
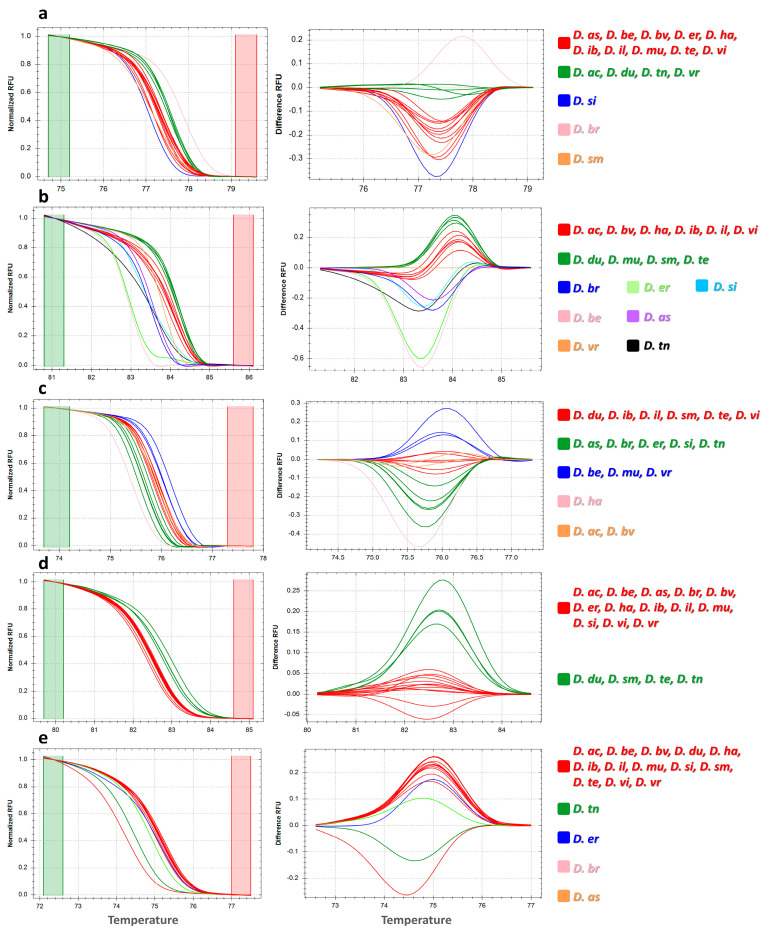
High-resolution melting profile showing normalized melting curves (left) and difference curves (right) for (**a**) HRM500, (**b**) ITS1, (**c**) *matk*, (**d**) *rbcl*, and (**e**) *trnL-F*. For acronyms, see Table 2 caption.

**Figure 4 genes-14-01594-f004:**
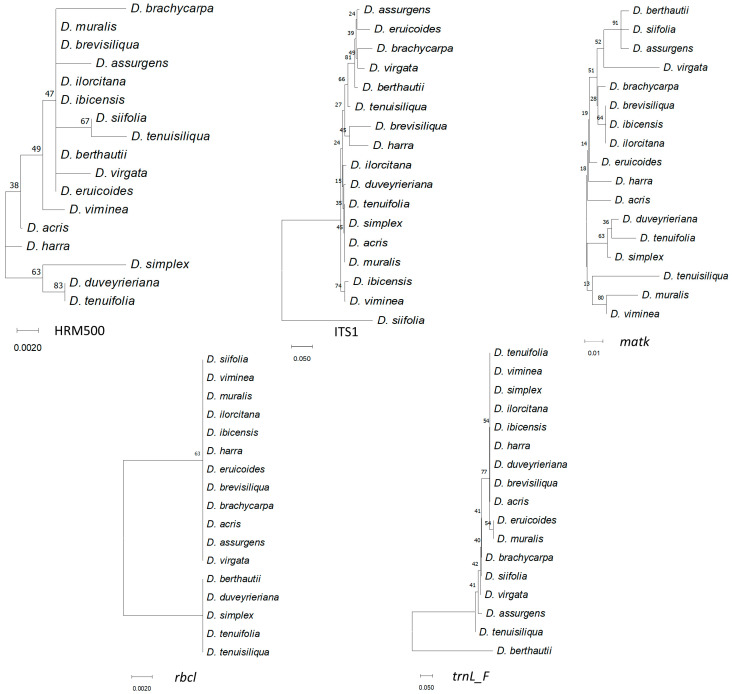
Phylogenetic analysis obtained from the five barcode sequences. The phylogenetic tree was inferred with 10,000 bootstraps using the neighbor-joining method. The evolutionary distances were computed using the Kimura 2-parameter model. Numbers above branches indicate the percentage of replicate trees, in which the associated taxa clustered together in the bootstrap test.

**Figure 5 genes-14-01594-f005:**
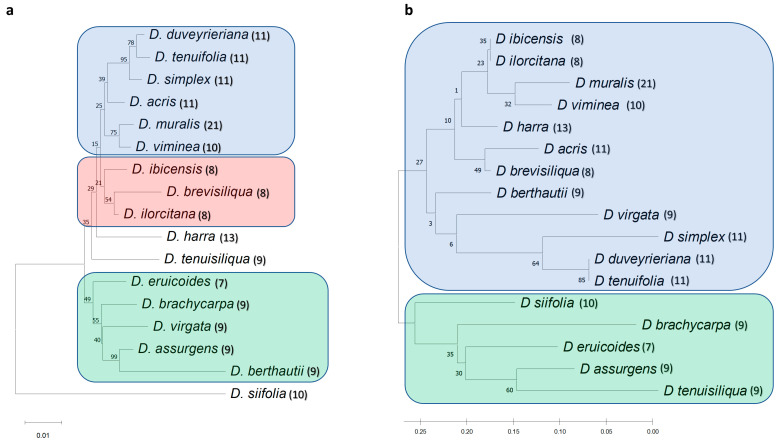
Phylogenetic analysis on combined polymorphisms from the five barcode regions. The phylogenetic tree was inferred by using the neighbor-joining method and the Kimura 2-parameter model. Bootstrap trees inferred from 10,000 replicates are shown: (**a**) Phylogenetic tree on concatenated barcode sequences; (**b**) Phylogenetic tree considering polymorphisms from HRM profiles. Numbers above branches indicate the percentage of replicate trees, in which the associated taxa clustered together in the bootstrap test. Phylogenetic analyses were conducted in MEGA X. In brackets are indicated the haploid chromosome numbers for each *Diplotaxis* species.

**Figure 6 genes-14-01594-f006:**
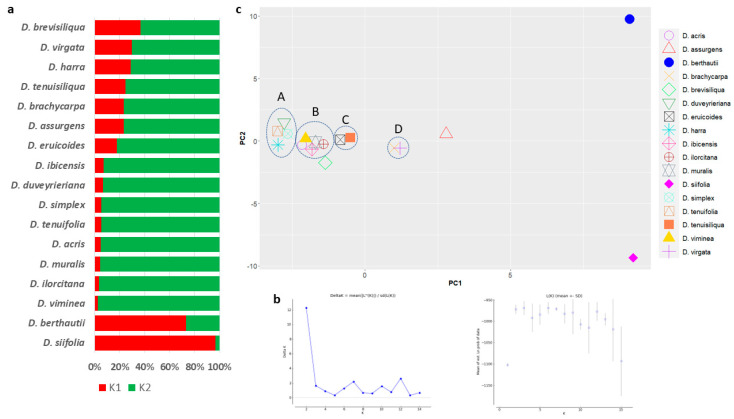
Population structure of 17 *Diplotaxis* spp. accessions based on 5 barcode regions. (**a**) STRUCTURE analysis considering K = 2 clusters. For each genotype, horizontal bars represent the allele frequency for each subpopulation (indicated in abscissis). (**b**) Evaluation of the best number of subpopulations within the Bayesian clustering analysis using Evanno’s method: on the right is the plot generated by STRUCTURE HARVESTER for the detection of the most likely number of clusters, the highest value was at K = 2, indicating that the 17 accessions likely form 2 sub-populations; on the left is the plot of mean likelihood L(K) and variance for 5 independent runs for each value of K for K = 1–15. (**c**) Loading plot in the first two components, showing the diversity of the collection.

**Table 1 genes-14-01594-t001:** Molecular markers for DNA barcoding used in the present study.

Marker Name	Foward Primer (5′–3′)	Reverse Primer (3′–5′)	Amplicon Size
ITS1	TTAGGCCGTGCGTATAGCTT	TTGCGTTCAAAGACTCGATG	249 bp
*trnL-F*	AGAAATTCCCGGTCCAAAAC	GGCCGTTACCGAAGTATCATT	107 bp
*rbcL*	CGGAGTTCCACCTGAAGAAG	TTGTAACGGTCAAGGCTGGT	105 bp
*matK*	TACGCCGCTTCTGATGAATA	TCTTTAGCCAACGACCCAAT	267 bp
HRM500	GATTCGAACCGTAGACCTGCTC	CCTTAAGGTGTAGCAAGTTTCA	115 bp

**Table 2 genes-14-01594-t002:** Nucleotide components of five nuclear and chloroplast regions for the 17 *Diplotaxis* species. For each marker, the total length of the aligned sequence is indicated in brackets.

	HRM500 (322)					ITS1 (274)					*matk* (241)					*rbcl* (73)					*trnL-F* (80)				
	Seq #	A	C	G	T	Seq	A	C	G	T	Seq	A	C	G	T	Seq	A	C	G	T	Seq	A	C	G	T
*D. ac* *	312	106	49	45	112	214	63	52	56	43	233	75	28	39	91	67	13	18	17	19	76	23	15	12	26
*D. as*	312	109	48	45	110	215	64	51	54	46	235	77	28	40	90	67	13	18	17	19	76	24	14	12	26
*D. be*	313	108	48	45	112	216	66	52	53	45	234	75	30	39	90	67	13	19	17	18	71	24	22	10	15
*D. br*	315	110	47	44	114	217	68	52	52	45	235	78	29	37	91	67	13	18	17	19	67	23	13	11	20
*D. bv*	313	106	48	45	114	216	62	53	57	44	234	76	31	37	90	69	14	18	17	20	80	23	18	12	27
*D. du*	313	105	49	47	112	217	66	52	55	44	235	76	30	38	91	68	14	19	17	18	76	23	15	12	26
*D. er*	313	108	48	45	112	218	68	51	53	46	234	76	29	38	91	68	14	18	17	19	78	23	17	12	26
*D. ha*	313	108	48	45	112	217	70	50	52	45	235	77	29	38	91	68	14	18	17	19	78	23	17	12	26
*D. ib*	313	108	48	45	112	214	67	52	52	43	236	77	31	38	90	70	14	20	17	19	78	24	16	12	26
*D. il*	314	108	48	45	113	213	63	51	56	43	236	77	31	38	90	68	14	18	17	19	79	23	17	12	27
*D. mu*	310	109	48	44	109	215	64	51	56	44	239	78	31	38	92	69	15	19	17	18	79	24	16	12	27
*D. si*	314	110	48	44	112	264	79	68	51	66	235	76	29	39	91	68	14	18	17	19	78	25	14	13	26
*D. sm*	314	104	49	47	114	214	63	51	56	44	235	76	31	38	90	68	14	19	17	18	76	23	15	12	26
*D. te*	312	104	49	47	112	216	65	51	56	44	236	77	29	40	90	68	14	19	17	18	77	24	15	12	26
*D. tn*	316	108	49	44	115	216	64	51	54	47	236	82	29	37	88	68	14	19	17	18	76	25	13	12	26
*D. vi*	309	109	47	43	110	217	66	52	55	44	236	77	31	39	89	68	14	18	17	19	78	24	16	12	26
*D. vr*	315	110	48	46	111	215	67	48	52	48	233	77	32	35	89	67	13	18	17	19	79	23	16	14	26

* Acronyms for the 17 assayed *Diplotaxis* species: *D. ac*, *D. acris*; *D. as*, *D. assurgens*; *D. be*, *D. berthautii*; *D. br*, *D. brachycarpa*; *D. bv*, *D. brevisiliqua*; *D. du*, *D. duveyrieriana*; *D. er*, *D. eruicoides*; *D. ha*, *D. harra*; *D. ib*, *D. ibicensis*; *D. il*, *D. ilorcitana*; *D. mu*, *D. muralis*; *D. si*, *D. siifolia*; *D. sm*, *D. simplex*; *D. te*, *D. tenuifolia*; *D. tn*, *D. tenuisiliqua*; *D. vi*, *D. viminea*; *D. vr*, *D. virgata*. # Seq = sequence length

**Table 3 genes-14-01594-t003:** Estimates of pairwise genetic distance (below the diagonal) and standard error estimate(s) (above the diagonal) within 17 Diplotaxis species. Species acronyms are reported in Table 2.

	*D. ac*	*D. as*	*D. be*	*D. br*	*D. bv*	*D. du*	*D. er*	*D. ha*	*D. ib*	*D. il*	*D. mu*	*D. si*	*D. sm*	*D. te*	*D. tn*	*D. vi*	*D. vr*
*D. ac*		0.005	0.007	0.005	0.005	0.004	0.005	0.005	0.004	0.003	0.004	0.009	0.004	0.004	0.005	0.004	0.006
*D. as*	0.025		0.006	0.004	0.006	0.006	0.005	0.006	0.005	0.005	0.006	0.009	0.006	0.006	0.006	0.006	0.005
*D. be*	0.049	0.032		0.007	0.008	0.007	0.007	0.008	0.007	0.007	0.008	0.010	0.008	0.008	0.008	0.008	0.007
*D. br*	0.026	0.018	0.042		0.006	0.006	0.005	0.006	0.006	0.005	0.006	0.009	0.006	0.006	0.006	0.006	0.005
*D. bv*	0.022	0.033	0.058	0.034		0.006	0.006	0.006	0.005	0.004	0.006	0.009	0.006	0.006	0.007	0.006	0.006
*D. du*	0.014	0.029	0.049	0.030	0.030		0.005	0.006	0.005	0.004	0.005	0.010	0.003	0.003	0.006	0.005	0.006
*D. er*	0.021	0.020	0.045	0.020	0.029	0.026		0.006	0.005	0.005	0.005	0.009	0.005	0.006	0.006	0.005	0.005
*D. ha*	0.026	0.035	0.059	0.035	0.032	0.029	0.032		0.005	0.005	0.006	0.010	0.006	0.006	0.006	0.005	0.006
*D. ib*	0.013	0.026	0.049	0.027	0.022	0.019	0.025	0.026		0.003	0.004	0.009	0.005	0.005	0.006	0.003	0.006
*D. il*	0.011	0.021	0.046	0.022	0.014	0.015	0.020	0.025	0.009		0.004	0.009	0.004	0.004	0.006	0.004	0.006
*D. mu*	0.014	0.027	0.052	0.028	0.028	0.019	0.023	0.030	0.018	0.013		0.010	0.004	0.004	0.005	0.003	0.006
*D. si*	0.083	0.080	0.103	0.083	0.084	0.088	0.084	0.091	0.078	0.078	0.085		0.010	0.010	0.010	0.009	0.010
*D. sm*	0.014	0.029	0.052	0.030	0.030	0.008	0.026	0.030	0.020	0.015	0.015	0.086		0.003	0.006	0.004	0.006
*D. te*	0.016	0.032	0.054	0.033	0.033	0.006	0.028	0.033	0.022	0.018	0.016	0.091	0.008		0.005	0.005	0.006
*D. tn*	0.026	0.029	0.052	0.030	0.039	0.030	0.030	0.036	0.032	0.027	0.026	0.092	0.028	0.026		0.006	0.006
*D. vi*	0.014	0.027	0.052	0.028	0.028	0.019	0.023	0.026	0.011	0.013	0.007	0.084	0.016	0.019	0.028		0.006
*D. vr*	0.030	0.020	0.045	0.022	0.035	0.035	0.025	0.036	0.032	0.027	0.028	0.087	0.033	0.035	0.032	0.028	

## Data Availability

All data generated in this study are available as Appendix A.

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
