# Peer review of "Application of High-Resolution Melting and DNA Barcoding for Discrimination and Taxonomy Definition of Rocket Salad (Diplotaxis spp.) Species"

_genes, 2023, doi:10.3390/genes14081594_

Round 1
Reviewer 1 Report
The MS revealed the genetic variability and phylogenetic relationships of the 17 Diplotaxis species with five barcode markers by HRM analysis. The MS presents good background literature research. The section Materials and methods were presented clear to enable other scientists to follow the results and continue recognition of other genetic doubts. The conclusions were derived from the obtained results during the performed studies. I suggest some compulsory revisions to improve the overall quality of the MS.
The genetic variability and phylogenetic relationships of the 17 Diplotaxis species can be more effectively revealed if the parameters of genetic diversity and genetic differentiation be supplemented in RESULTS
The chart format should be standardized according to the journal requirements. For example, the ruler is missing in Figure 1; the unit should be put into the title row, and the primer sequence direction should be added in Table 1.
The Latin and gene names need to be italicized in the whole MS.
In line144, 166 and other places, L should be μL.
Minor editing of English language required
Author Response
I would like to thank the reviewer for the attention to this work and for meaningful suggestions. The manuscript has been revised accordingly all changes respect to the previous version are in blue font. Below point by point response to comments:
1) The MS revealed the genetic variability and phylogenetic relationships of the 17 Diplotaxis species with five barcode markers by HRM analysis. The MS presents good background literature research. The section Materials and methods were presented clear to enable other scientists to follow the results and continue recognition of other genetic doubts. The conclusions were derived from the obtained results during the performed studies. I suggest some compulsory revisions to improve the overall quality of the MS.
#Thanks for the positive comment.
2) The genetic variability and phylogenetic relationships of the 17 Diplotaxis species can be more effectively revealed if the parameters of genetic diversity and genetic differentiation be supplemented in RESULTS
#The pairwise genetic distance among all the studied accessions is provided. A new Table (Table 3) is included aloing with comments in the results (L233-237)
3) The chart format should be standardized according to the journal requirements. For example, the ruler is missing in Figure 1; the unit should be put into the title row, and the primer sequence direction should be added in Table 1.
#For figure 1, unfortunatley i didn't placed the ruler when i took the photos. The figure 1 has been included in the manuscript just to show the diversity in terms of leaf shape. Photos have been taken with a camera mounted on a stand, so all images are to the same scale. I added a comment at the end of the caption
#For formatting reasons, Table 1 has been modified. I've removed those column displaying information already reported in the text (Region, Organelle, GenBank Accession No.,Sequence lenght) and left only the new information. The table is thus easier to read,
4) The Latin and gene names need to be italicized in the whole MS.
In line144, 166 and other places, L should be μL.
# The manuscript has been careful reviewed all typos and format errores have been corrected
I hope all answers satisfy the reviewer concerns, i'm available for any additional revisions if required
Kind regards
Reviewer 2 Report
The manuscript described the first investigation of Diplotaxis species using newly developed barcode markers that integrate DNA sequencing and high-resolution melting analysis.
The phylogenetic relationships between the 17 tested species validated previous inferences based on morphological, biochemical, and molecular data, indicating the reliability of the tested marker.
The findings improved the resolution of the evolutionary distance between D. siifolia and the Diplotaxis gene pool. ITS1 has been identified as the barcoding sequence with the maximum discriminatory power for identifying species.
The DNA barcode and HRM demonstrate the efficacy of the approach utilised in this study for identifying species and analysing genetic relationships.
Setting up research using multiple analyzes represents a more comprehensive approach to distinguishing species. The results of the work contribute to new knowledge in the field of DNA barcoding and species identification. I suggest publishing the work in this form.
Author Response
I would like to thank the reviewer for reading the the work, thanking for the positive comments. I hope this manuscript would be useful for the community working with Diplotaxis species.
Kind regards